# Political context and immigrants' work-related performance errors: Insights from the National Basketball Association

**Benjamin A. Korman**[1,2]*, **Florian Kunze**[1,2]

**1** Chair of Organizational Behavior, University of Konstanz, Konstanz, Germany, **2** Cluster of Excellence "The Politics of Inequality", University of Konstanz, Konstanz, Germany

* benjamin.korman@uni-bamberg.de

**Data Availability Statement:** The data and code used for the analyses in this study are publicly available on the Open Science Framework platform (https://osf.io/kex6j/?view_only=ae9b627a921d493683e9b53ee19af0ad).

## Abstract

In numerous countries, both international migration and regional support for far-right political parties are on the rise. This is important considering that a frequent aim of far-right political parties is to aggressively limit the inflow of immigrants. Understanding how regional far-right political support affects the immigrants working in these regions is therefore vital for executives and organizations as a whole. Integrating political science research at the macro-level with stereotype threat theory at the individual level, we argue that regional far-right political support makes negative immigrant stereotypes salient, increasing the number of work-related performance errors conducted by immigrants while reducing those by natives. Using objective field data from a professional sports context, we demonstrate how subordinates' immigrant status interacts with the political context in which they reside to predict their frequency of performance errors.

## Introduction

International migration has been increasing steadily for decades and an estimated 405 million people are predicted to live outside their country of birth by 2050 [1]. With this development has come a political polarization process [2] that has garnered public support for far-right political parties [3]. Recent European examples of this include the gains by Marine Le Penn and the National Rally in France as well as Viktor Orban and Fidesz in Hungary. In the United States (U.S.), our study's political context, this includes the intense and, at times, violent support of the Republican party under Donald Trump. These examples suggest that regional far-right political support has the power to shape the social context of the regions in which immigrants reside.

Studies in economics and political science have shown that immigrants face greater prejudice and discrimination if residing in areas where the political environment is heavily influenced by far-right political parties. For instance, increasing public support for far-right political parties relates to lower life satisfaction in immigrants [4] and their increased perceptions of discrimination [5]. Relatedly, experimental results have shown anti-immigrant propaganda (which is often used by far-right political parties) to reduce the intellectual performance

**Funding:** The authors (BAK & FK) acknowledge funding from the Deutsche Forschungsgemeinschaft (DFG – German Research Foundation: [https://www.dfg.de] under Germany's Excellence Strategy – EXC-2035/1 – 390681379. The funders had no role in the study design, data collection and analysis, decision to publish, or preparation of the manuscript.

**Competing interests:** The authors have declared that no competing interests exist.

of adolescent immigrants [6]. Despite these findings, and despite calls for management researchers to incorporate the external context into the study of immigrants in organizations [7,8], little work has investigated when far-right politics seeps into organizations or how this affects immigrant employees. This is worrying as organizations can play a pivotal role in the successful integration of immigrants by helping them come into contact with natives [9], acquire useful cultural skills [10], and develop a strong identity with their host country [11].

Using stereotype threat [12,13] as our theoretical framework, we call attention to the role of the regional political climate in which immigrants reside (i.e., regional far-right political support) in predicting immigrants' work-related performance errors. Specifically, we propose that regional far-right political support increases immigrants' awareness that they may be judged based on negative stereotypes of their immigrant status [14], in turn causing them to focus their attention on otherwise well-learned and automatic tasks, resulting in increased errors when performing these tasks. We argue that this can be costly for immigrants as increased work-related performance errors can put their employment, and thus financial stability, at risk.

With this study, we contribute to the management literature on immigrants, an important yet largely ignored group of the working population [15,16]. Specifically, we demonstrate how even the most skilled and successful immigrants (e.g., professional athletes in the most competitive basketball league worldwide) can show increased performance errors linked to the regional far-right political support where they reside. This work therefore counterbalances previous studies that commonly focused on immigrants in blue-collar positions [17,18] to shed light on the experiences of immigrants at (or near) the peak of their careers. Furthermore, we build on macro-level findings in the fields of economics [4] and political science [19] to show how regional far-right political support can affect immigrants' work-related outcomes at the individual-level, thereby answering calls for increased focus on the importance of context in organizational research [20,21]. Our findings suggest that regional far-right political support is associated with increased work-related performance errors by immigrant workers (Fig 1), a negative outcome for their organizations. Finally, our work demonstrates the value of using the professional sports context to study immigrants' experiences within the workplace. The professional sports industry offers large, openly available datasets with objective measures of athletes' real-world behavior nested within teams, frequently including information regarding

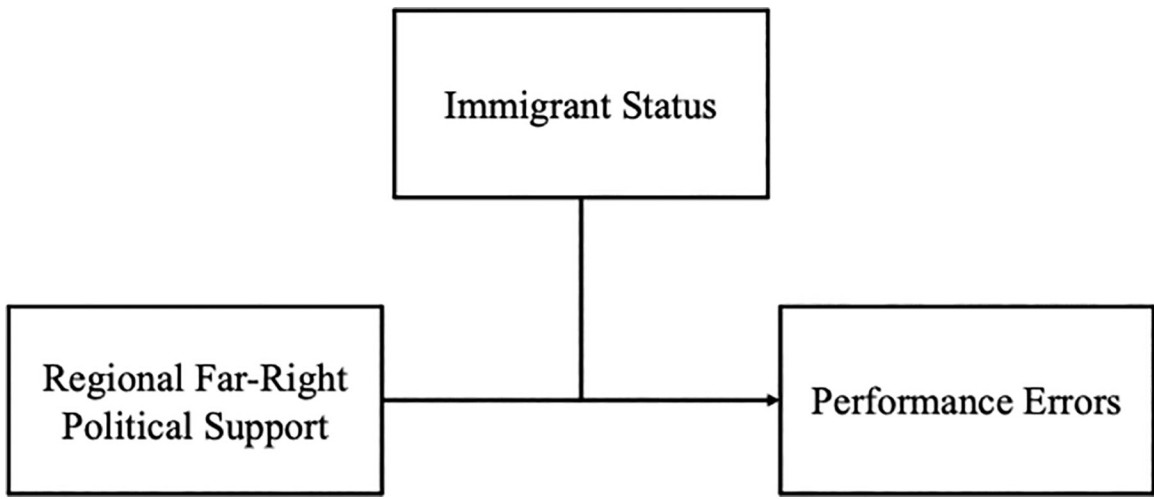

**Fig 1. Theoretical model.**

athletes' place of birth. As such, this data source can be a boon for immigration research due to the difficulty with finding appropriately-sized immigrant samples within organizations [7]. In sum, we demonstrate that a threatening political environment is associated with negative work-related outcomes for even the most skilled and successful immigrants in the unique organizational context offered by the National Basketball Association (NBA).

## Theory and hypotheses

Immigrants are often the targets of stereotyping and prejudice by political parties and their leaders. This has recently been the case in the U.S., the political context of our study, where former president Donald Trump–leader of the Republican party–consistently implemented, or attempted to implement, far-right and anti-immigrant policy [22] while casting immigrants in a negative light [23]. This kind of far-right political rhetoric has been associated with greater anti-immigrant attitudes in the general population [24] as well as the mobilization of those with such sentiments [25], and it is no surprise that immigrants are often discriminated against by natives in their host country [18]. Regional far-right political support therefore likely exacerbates immigrants' perceptions of having outsider status, with potential consequences for their work performance. In line with this regional perspective on organizational behavior, Harrison and colleagues [7] advised researchers to investigate how the political-cultural context of the regions where immigrants work affects their work-related outcomes. Based on their recommendation, we will explore how the regional political climate affects immigrants' work-related performance errors in the context of professional sports.

According to stereotype threat theory [12,13], individuals experience stereotype threat when they are concerned about confirming or being reduced to a negative stereotype about a group to which they belong to. Stereotypes can relate to individuals' country of origin [26] and immigrants are often claimed to be one of the target groups of stereotype threat [16]. Negative stereotypes can elicit a sense of uncertainty which, in turn, decreases performance [27]. This is because uncertainty can lead to increased attention toward threat-related cues in the environment and increased monitoring of one's own performance [27–29]. This *explicit monitoring* is thought to direct attention to well-learned and automatic tasks [26], thus disrupting their otherwise fluent execution [30–32]. Furthermore, chronic stereotype threat (i.e., stereotype threat experienced over a long period of time) has been associated with decreased working memory [33] and decreased confidence in attaining one's goals [34,35], both of which are important for performance.

Stereotype threat is particularly pernicious as it can affect one's performance even if one does not believe the stereotype [36], is not aware that a stereotype has even been activated [37], or has not actually been stereotyped [38]. Furthermore, stereotype threat theory proposes that individuals invested in their performance domain (e.g., employees) are most affected by stereotype threat [13,39].

## Stereotype threat and regional far-right political support

Stereotype threat occurs when the minority identity of an individual is activated and this can be elicited by various cues [31]. Social interactions can be a source of such cues [40], but exposure to devaluing content in the media can also impair negatively stereotyped groups [6]. Immigrants who reside in regions where devaluing content is more prevalent, such as in regions with higher far-right political support, are therefore more likely to experience stereotype threat-eliciting cues when engaging with locals or consuming various forms of local media (e.g., social media, television, newspapers, public radio). In line with this, previous research has found that newspaper coverage of professional athletes is biased depending on

players' nationality [41] and ethnic group membership [42]. Even without direct insight into locals' anti-immigrant attitudes, subtle yet symbolic cues such as apparel (e.g., the red MAGA hats supporting Donald Trump) [43], election posters, and locally individualized ads and discussions in television and social media may inform immigrants whether the social environment of the region where they reside is potentially threatening or not. Taken together, we expect that sustained exposure to negative cues reflective of regional far-right political support in the region where immigrants reside will lead immigrants to experience chronic stereotype threat, increasing their work-related performance errors. In contrast, immigrants living in regions where these cues are less common (i.e., regions with less far-right political support), should be less affected. Therefore, we propose the following:

*Hypothesis 1*: Regional far-right political support is positively associated with performance errors in immigrant employees.

## Stereotype lift and regional far-right political support

Social interactions, media, and symbolic cues devaluing immigrants may not only affect immigrants' performance errors but could also have the potential to affect native employees. Previous research has shown that non-stereotyped individuals (i.e., members of the ingroup) demonstrate increased performance when an outgroup is negatively stereotyped [44–47]. This phenomenon is known as *stereotype lift* and it has been theorized to boost performance (i.e., decrease performance errors) by increasing individuals' downward social comparisons with the denigrated outgroup [47], in turn increasing their sense of self-efficacy and self-worth [48]. As a result, ingroup members (i.e., natives) may be less prone to "self-doubt, anxiety, and fear of rejection that could otherwise hamper performance" [47, p. 457]. Empirical evidence for this comes from studies showing that stereotype lift effects on a motor task were mediated by increased self-confidence and task involvement [45], as well as decreased anxiety [49]. Taking this previous work into consideration, we expect that sustained exposure to cues reflective of far-right political support in the region where natives reside will lead natives to experience stereotype lift, decreasing their work-related performance errors. In contrast, natives residing in regions where these cues are less common (i.e., regions where far-right political support is lower), should be less affected. We therefore hypothesize the following:

*Hypothesis 2*: Regional far-right political support is negatively associated with performance errors in native employees.

# Materials and methods

## Data and sample

The sample size was determined by the number of basketball players playing in the 2020–2021 regular season of the NBA. We specifically selected the 2020–2021 season for this study as it spans the transfer of presidential power in the U.S. following the 2020 election and thus covers a time in which political affiliation and support for Donald Trump and his Republican party were both visible and polarizing. Of note, the 2020–2021 regular season of the NBA went from December 22nd, 2020 to May 16th, 2021 and supporters of Donald Trump stormed the nation's capital on January 6th, 2021 in an attempt to overturn the results of the presidential election in which he lost.

Data were collected for all five hundred and forty (*N* = 540; 100% male) players from the official NBA website (www.nba.com). Furthermore, we matched the NBA data with presidential election data at the state level openly available from the Harvard University dataverse [50]. Matching of players to state election data was based on the home location of their team and 522 players were matched to 23 regions. One team consisting of 18 players could not be

matched to the presidential election data because the team was located in Toronto, Canada. Teams consisted on average of 18.2 players (*SD* = 1.9). Players in our final sample were on average 26.0 years old (*SD* = 4.1) and came from 41 different countries, the most coming from the U.S. (77%), Canada (3%), and France (2%). Of those with immigrant status, 48.3% were born in Europe, 5.9% in Asia, 13.6% in Africa, 5.9% in South America, 18.6% in North America, and 7.6% in Australia/Oceania.

## Measures

**Immigrant status.**   Similar to other research on immigrants as a focal group [51–53], immigrant status was based on players' birthplace. Specifically, players' immigrant status was dummy coded with 0 representing non-immigrant status (i.e., born in the U.S.) and 1 representing immigrant status (i.e., born outside the U.S.). Information regarding players' birthplace was obtained from their official NBA profile pages. One hundred and eighteen players (23%) had immigrant status.

**Regional far-right political support.**   We used the percentage of votes for Donald Trump in the 2020 presidential election as a proxy for regional far-right political support in each players' region of residence. According to data from the Manifesto Project (https://manifesto-project.wzb.eu/), an international project analyzing the content of political parties' electoral manifestos, the Republican party's manifesto under Donald Trump was estimated as being farther right than that of well-known far-right parties in Europe such as the National Rally in France and the Austrian Freedom Party [54]. Importantly, anti-immigrant sentiment was found to be a major determinant of Donald Trump voter support [55]. Furthermore, Republicans' dissatisfaction with immigration levels in the U.S. grew to 87% after Donald Trump left office, indicating their strong support of his immigration policies [56]. Overall, Donald Trump received 47% of the vote in the 2020 presidential election. In our sample, players worked in states that had on average 45.7% (*SD* = 11.6) regional far-right political support, with this support ranging from 5.4% to 65.4% across regions.

**Performance errors.**   Players' performance errors were operationalized at the individual level by the number of times that they personally lost possession of the basketball to the opposing team before a shot was attempted (i.e., committed a "turnover"). Examples of this include stepping out of bounds with the ball or having the ball stolen by an opponent. This "turnover" measure is a relevant metric to assess players' performance errors and is therefore regularly monitored by coaches and teams and has been used in multiple studies on basketball performance [57].

**Covariates.**   To account for the fact that performance errors (as operationalized above) can only occur when a player has possession of the ball, we accounted for players' total number of ball possessions. Furthermore, because some forms of performance error result from having possession of the ball for too long (e.g., a "shot clock" violation), we controlled for the total amount of time (in minutes) players had possession of the ball. We also accounted for players' age and position played as their basketball experience and role within the team may determine their number of performance errors. In addition, and as recommended in the literature on stereotype threat [38], we accounted for factors reflecting how players are treated by their organizations. These covariates included players' salary and the total number of minutes they were allowed to play. Players' skin color was also included as a covariate to show that immigrant status accounts for variance in performance errors above that of skin color. To measure player's skin color, the profile photo of each player on the official NBA website was coded by two research assistants blind to the focus of the study using the single-item Fitzpatrick Skin Phototype Scale [58]. This measure involves a 6-point Likert scale (1 = 'light, pale white', 6 = 'black,

very dark brown to black') accompanying a colored image with an example skin color reflecting each scale option. As recommended [59], interrater reliability was tested using two-way random-effects intraclass correlation coefficients (ICCs). Interrater reliability for the variable skin color was found to be high ICC(A,1) = .97, validating its use in our analysis. We also controlled for team-level variables that were considered likely to affect players' number of performance errors. These included a measure of how successful the team was during the season (as indicated by the number of games won in the regular season) and a binary variable indicating whether the team started the season with a new head coach (nine teams did). We included this as previous research has shown that coaching changes can affect team performance [60,61]. Lastly, and has been recommended [62], we included the cluster (i.e., level-2) means of our individual-level (level-1) predictors (i.e., immigrant status, number of ball possessions, time with ball possession, age, salary, minutes played, and skin color) to account for these variables at the team level.

All variables related to game-play represent statistics totaled per player across the entire 2020–2021 regular season in which each team played 72 games. The means, standard deviations, and zero-order correlations among all study variables are presented in Table 1.

## Results

Due to the clustered nature of the data (i.e., players clustered within teams), we used a correlated random effects (CRE) approach and conducted regression analyses with cluster robust-standard errors (CR-SEs) as has been recommended [62,63]. This approach offers more consistent estimates and allows us to test for both within (level-1) and between (level-2) effects [64]. An additional advantage of clustered standard errors is that fewer assumptions about the appropriate specification of random effects at higher levels (e.g., the team level) are needed [65]. To test Hypotheses 1 and 2, we set regional far-right political support as the independent variable, immigrant status as the moderator, and performance errors as the dependent variable. The following were included in our main analysis as covariates: Players' number of ball possessions, amount of time with ball possession, age, position, salary, minutes of gameplay, skin tone, and the cluster mean of each of these level-1 variables. Included were also team performance and coaching changes at the team level and the data were analyzed using Stata/SE 16.1 [66].

As expected, regional far-right political support interacted with immigrant status to predict players' performance errors, B = 0.42, $p < .001$ (see Table 2: Model 1 for details). Follow-up simple slopes analysis show that regional far-right political support is positively related to performance errors in immigrant players, B = 0.245, $p < .001$, and negatively related to performance errors in native players, B = -0.178, $p = .001$ (Fig 2). Based on these estimates, immigrants in our sample executed on average 5.7 (11%) more performance errors when residing in regions higher (+1 SD) in regional far-right political support compared to immigrants residing in regions lower (-1 SD) in such support. These findings support Hypothesis 1. Furthermore, our results suggest that native players in our sample executed on average 4.1 (8%) less performance errors when residing in regions higher (+1 SD) in regional far-right political support compared to natives residing in regions lower (-1 SD) in such support. These findings support Hypothesis 2.

### Robustness checks and supplemental analysis

To establish the robustness of our findings, we retested Hypothesis 1 and 2 using a simplified model. Players' total number of ball possessions was kept in our robustness check analysis because, as previously noted, players cannot demonstrate performance errors without

**Table 1. Means, standard deviations, and zero-order correlations among variables.**

| Variables | M | SD | (1) | (2) | (3) | (4) | (5) | (6) | (7) | (8) | (9) | (10) | (11) |
|---|---|---|---|---|---|---|---|---|---|---|---|---|---|
| (1) Performance errors | 53.08 | 51.47 | - | | | | | | | | | | |
| (2) Immigrant status | 0.23 | 0.42 | 0.038 | - | | | | | | | | | |
| (3) Regional far-right political support | 45.72 | 11.60 | -0.027 | 0.031 | - | | | | | | | | |
| (4) Ball possessions | 1620.83 | 1369.64 | 0.936 | 0.033 | -0.019 | - | | | | | | | |
| (5) Time with ball possession | 83.09 | 99.82 | 0.874 | -0.019 | -0.013 | 0.884 | - | | | | | | |
| (6) Age | 25.99 | 4.15 | 0.118 | -0.026 | -0.074 | 0.173 | 0.122 | - | | | | | |
| (7) Position (Center) | 0.19 | 0.39 | -0.064 | 0.151 | -0.037 | -0.067 | -0.131 | 0.053 | - | | | | |
| (8) Position (Power forward) | 0.21 | 0.41 | -0.030 | 0.072 | -0.021 | -0.031 | -0.101 | 0.025 | -0.247 | - | | | |
| (9) Position (Power guard) | 0.19 | 0.39 | 0.126 | -0.089 | -0.014 | 0.126 | 0.271 | -0.016 | -0.234 | -0.250 | - | | |
| (10) Position (Shooting forward) | 0.18 | 0.38 | -0.083 | 0.002 | 0.054 | -0.079 | -0.098 | -0.050 | -0.222 | -0.238 | -0.225 | - | |
| (11) Position (Shooting guard) | 0.24 | 0.43 | 0.045 | -0.127 | 0.019 | 0.044 | 0.053 | -0.012 | -0.267 | -0.285 | -0.270 | -0.257 | - |
| (12) Salary | 7004467 | 9026380 | -0.074 | 0.068 | 0.007 | -0.082 | -0.076 | -0.037 | 0.017 | 0.016 | 0.007 | -0.022 | -0.018 |
| (13) Minutes played | 964.16 | 699.50 | 0.842 | 0.016 | -0.027 | 0.924 | 0.715 | 0.182 | -0.082 | -0.012 | 0.024 | -0.043 | 0.103 |
| (14) Skin color | 4.17 | 1.48 | 0.011 | -0.318 | -0.038 | -0.019 | 0.032 | -0.032 | -0.016 | 0.089 | 0.018 | -0.064 | -0.029 |
| (15) New head coach | 0.33 | 0.47 | -0.019 | -0.014 | 0.143 | -0.024 | -0.025 | 0.049 | -0.010 | 0.035 | -0.068 | 0.033 | 0.009 |
| (16) Team performance | 36.03 | 10.11 | 0.035 | -0.021 | -0.194 | 0.067 | 0.048 | 0.217 | -0.013 | 0.048 | -0.038 | -0.006 | 0.007 |
| (17) Immigrant status (CM) | 0.23 | 0.09 | 0.040 | 0.220 | 0.143 | 0.023 | 0.010 | -0.052 | 0.016 | 0.016 | -0.013 | -0.004 | -0.014 |
| (18) Ball possessions (CM) | 1620.83 | 203.24 | 0.123 | 0.034 | -0.126 | 0.148 | 0.093 | 0.049 | 0.014 | 0.011 | 0.034 | -0.027 | -0.031 |
| (19) Time with ball possession (CM) | 83.09 | 10.31 | 0.098 | 0.021 | -0.131 | 0.134 | 0.103 | 0.055 | 0.036 | 0.002 | 0.011 | -0.010 | -0.037 |
| (20) Age (CM) | 25.99 | 1.34 | 0.016 | -0.035 | -0.229 | 0.022 | 0.017 | 0.323 | -0.025 | 0.043 | -0.047 | 0.011 | 0.015 |
| (21) Salary (CM) | 7004467 | 2503235 | 0.003 | 0.022 | 0.025 | 0.003 | 0.003 | 0.018 | -0.048 | 0.025 | -0.015 | -0.029 | 0.059 |
| (22) Minutes played (CM) | 964.16 | 111.57 | 0.111 | 0.039 | -0.167 | 0.140 | 0.098 | 0.049 | 0.027 | -0.011 | 0.020 | -0.006 | -0.028 |
| (23) Skin color (CM | 4.17 | 0.42 | -0.009 | -0.138 | -0.137 | -0.001 | 0.008 | -0.036 | 0.023 | -0.032 | 0.017 | 0.034 | -0.038 |

| Variables | (12) | (13) | (14) | (15) | (16) | (17) | (18) | (19) | (20) | (21) | (22) | (23) |
|---|---|---|---|---|---|---|---|---|---|---|---|---|
| (12) Salary | - | | | | | | | | | | | |
| (13) Minutes played | -0.067 | - | | | | | | | | | | |
| (14) Skin color | -0.059 | -0.005 | - | | | | | | | | | |
| (15) New head coach | 0.036 | -0.044 | 0.067 | - | | | | | | | | |
| (16) Team performance | 0.022 | 0.075 | -0.026 | -0.034 | - | | | | | | | |
| (17) Immigrant status (CM) | 0.028 | 0.028 | -0.177 | -0.064 | -0.094 | - | | | | | | |
| (18) Ball possessions (CM) | 0.006 | 0.150 | -0.001 | -0.164 | 0.451 | 0.154 | - | | | | | |
| (19) Time with ball possession (CM) | 0.009 | 0.152 | 0.021 | -0.243 | 0.468 | 0.096 | 0.905 | - | | | | |
| (20) Age (CM) | 0.016 | 0.024 | -0.032 | 0.152 | 0.672 | -0.161 | 0.151 | 0.169 | - | | | |
| (21) Salary (CM) | 0.277 | 0.014 | -0.088 | 0.129 | 0.081 | 0.101 | 0.022 | 0.033 | 0.057 | - | | |
| (22) Minutes played (CM) | 0.023 | 0.159 | -0.002 | -0.274 | 0.473 | 0.177 | 0.941 | 0.951 | 0.151 | 0.085 | - | |
| (23) Skin color (CM) | -0.087 | -0.001 | 0.282 | 0.237 | -0.092 | -0.630 | -0.004 | 0.076 | -0.112 | -0.313 | -0.007 | - |

*Notes*. N = 522 players in 29 teams. CM = cluster mean. Zero-order correlations less than -0.088 or greater than 0.088 are significant at the .05 level.

possession of the ball. Furthermore, and as in our previous analysis, we included the cluster (i.e., level-2) means of our individual-level (level-1) predictors (i.e., immigrant status and number of ball possessions) to account for these variables at the team level.

Using the same procedure as in our main analysis, we again found that regional far-right political support interacted with immigrant status to predict players' performance errors, B = 0.423, $p < .001$ (see Table 2: Model 2 for details). Furthermore, and similar to the results of our main analysis, follow-up simple slopes analyses show that regional far-right political support is positively related to performance errors in immigrant players, B = 0.222, $p = .002$, and

**Table 2. Regression results.**

| Variables | Model 1 | | | Model 2 | | | Model 3 | | |
|---|---|---|---|---|---|---|---|---|---|
| | B | SE | p | B | SE | p | B | SE | p |
| Immigrant status (0 = native; 1 = immigrant) | -17.511 | 4.383 | < .001 | -19.167 | 4.920 | .001 | -22.766 | 12.568 | .081 |
| Regional far-right political support | -0.178 | 0.049 | .001 | -0.202 | 0.047 | < .001 | -0.088 | 0.214 | .685 |
| Immigrant status × Regional far-right political support | 0.423 | 0.088 | < .001 | 0.423 | 0.092 | < .001 | 0.565 | 0.262 | .040 |
| Immigrant status × Skin color | | | | | | | 2.178 | 3.388 | .526 |
| Regional far-right political support × Skin color | | | | | | | -0.021 | 0.051 | .676 |
| Immigrant status × Skin color × Regional far-right political support | | | | | | | -0.055 | 0.078 | .485 |
| Ball possessions | 0.026 | 0.005 | < .001 | 0.035 | 0.001 | < .001 | 0.026 | 0.005 | < .001 |
| Time with ball possession | 0.129 | 0.041 | .004 | | | | 0.131 | 0.041 | .003 |
| Age | -0.544 | 0.195 | .009 | | | | -0.552 | 0.202 | .011 |
| Position | | | | | | | | | |
| Power forward | -0.747 | 2.779 | .790 | | | | -0.965 | 2.729 | .726 |
| Power guard | -5.725 | 2.525 | .031 | | | | -5.855 | 2.580 | .031 |
| Shooting forward | -2.270 | 2.755 | .417 | | | | -2.558 | 2.807 | .370 |
| Shooting guard | -2.360 | 2.576 | .367 | | | | -2.517 | 2.637 | .348 |
| Salary | 0.000 | 0.000 | .905 | | | | 0.000 | 0.000 | .944 |
| Minutes played | 0.002 | 0.005 | .718 | | | | 0.002 | 0.005 | .675 |
| Skin color | 0.875 | 0.512 | .099 | | | | 1.974 | 2.341 | .406 |
| New head coach (0 = no; 1 = yes) | -1.648 | 1.132 | .156 | | | | -1.457 | 1.231 | .247 |
| Team performance | -0.202 | 0.047 | < .001 | | | | -0.204 | 0.049 | < .001 |
| Immigrant status (CM) | 14.162 | 8.111 | .092 | 14.585 | 8.974 | .115 | 14.436 | 8.163 | .088 |
| Ball possessions (CM) | 0.021 | 0.008 | .020 | -0.006 | 0.003 | .023 | 0.020 | 0.008 | .021 |
| Time with ball possession (CM) | -0.623 | 0.168 | .001 | | | | -0.637 | 0.173 | .001 |
| Age (CM) | 1.714 | 0.506 | .002 | | | | 1.697 | 0.529 | .003 |
| Salary (CM) | 0.000 | 0.000 | .633 | | | | 0.000 | 0.000 | .771 |
| Minutes played (CM) | 0.013 | 0.018 | .469 | | | | 0.015 | 0.018 | .419 |
| Skin color (CM) | 1.291 | 1.451 | .381 | | | | 1.025 | 1.427 | .478 |
| Constant | -22.356 | 14.943 | .146 | 11.542 | 3.743 | .005 | -24.945 | 18.362 | .185 |

*Notes*. N = 522 players in 29 teams. Model 1 = Main analysis; Model 2 = Robustness check analysis; Model 3 = Supplementary analysis; CM = cluster mean. The effects of position were estimated using "center" as the comparison group.

negatively related to performance errors in native players, B = -0.202, $p < .001$. These findings provide additional support for Hypotheses 1 and 2. To further test the robustness of our findings, we re-ran our main analysis and robustness check analysis while excluding the 10% of players with the fewest number of ball possessions throughout the season (i.e., less than 115). This was to rule out that our findings were driven by players with few opportunities for performance errors. Exclusion of these players ($n = 53$) did not significantly affect our results or our interpretation of them and, therefore, they are not discussed further.

In an attempt to better understand the negative relationship between regional far-right political support and immigrants' performance errors, we tested a model in which skin color was allowed to interact with regional far-right political support and immigrant status. This was done to determine whether immigrants with darker skin color exhibit greater or fewer performance errors relative to those with lighter skin color in regions with higher regional far-right political support. Using the same procedure as in our main analysis while including the 3-way

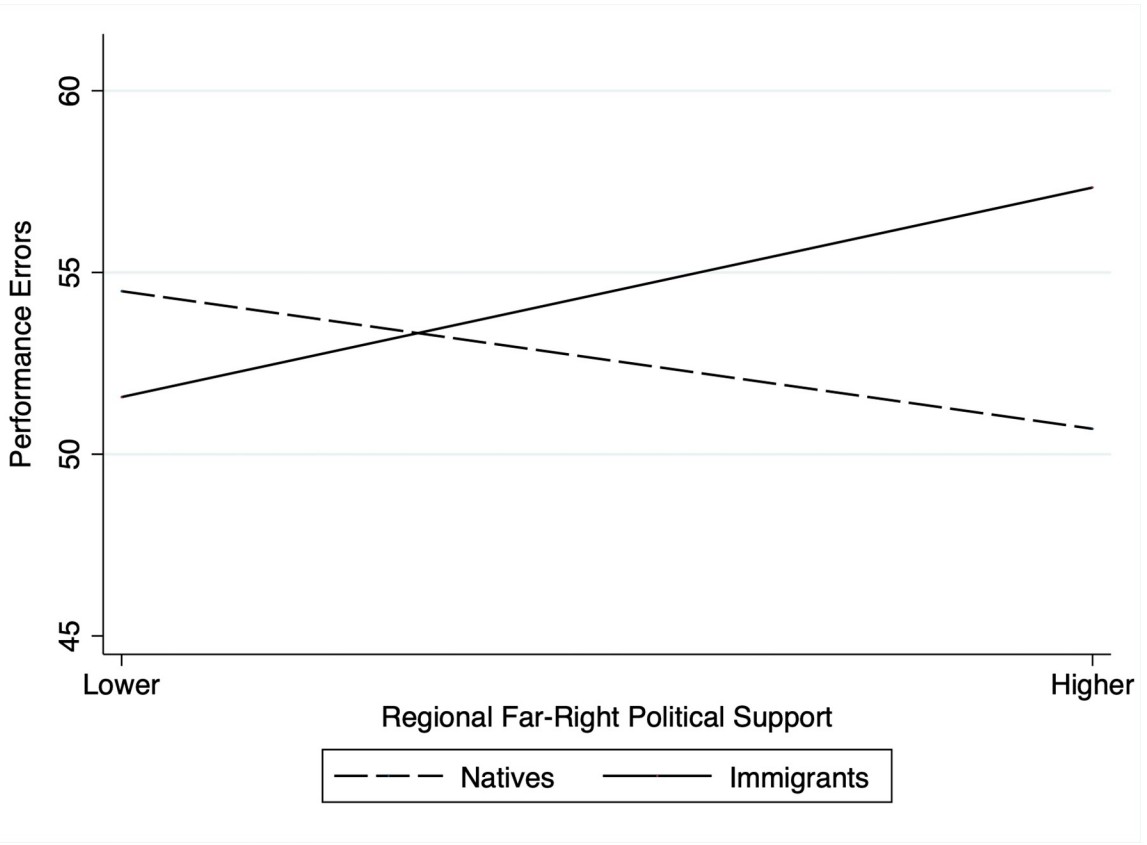

**Fig 2. Regional far-right political support × immigrant status interaction effect on performance errors.**

interaction between regional far-right political support, immigrant status, and skin color, we failed to find statistically significant differences between indviduals based on their performance errors, B = -0.055, *p* = .485 (see Table 2: Model 3 for details). This lack of a finding suggests that immigrants residing in regions with higher far-right political support exhibit more performance errors, regardless of their skin color.

## Discussion

Immigrants, along with the role of the external environment in predicting work-related outcomes, have been largely ignored in management research [7,15]. We contribute to these topics by using objective data from the NBA combined with U.S. presidential election data to show how the political environment of the region where immigrants reside affects their work-related performance errors. Our results provide real-world evidence that immigrants residing in regions with higher far-right political support execute significantly more performance errors than those residing in regions with lower far-right political support, regardless of their skin color. Moreover, our results also suggest that natives make fewer performance errors when residing in regions with higher far-right political support. This suggests that differences in performance errors between immigrants and natives may result from a combination of stereotype threat and stereotype lift effects, further emphasizing the need to take environmental factors into account when assessing immigrants' work-related outcomes relative to their native colleagues. Together, our findings highlight important relationships linking the external political environment with the work-related performance of organizations' immigrant (as well as

native) employees and we introduce professional sports as a unique and valuable organizational context in which to test such phenomena.

## Theoretical contributions

Our study makes several theoretical contributions to research on immigrants in organizations. First, although immigrants make up an ever-growing subgroup of the workforce, they have received little attention in the organizational behavior and management literatures [7]. Furthermore, the limited research that has been conducted on the individual-level work-related outcomes of immigrants has focused largely on blue-collar workers [17,18]. By focusing on immigrants widely recognized as among the best in their profession, we show under what circumstances, and to what extent, even the most skilled and successful immigrants demonstrate increased work-related performance errors. In so doing, we extend the current literature on what has been considered an "invisible" group [67] by highlighting the experiences of its most visible members.

Second, our work complements research on immigrants at the macro-level that has found associations between regional far-right political support and psychological variables such as lower life satisfaction [4] and higher perceived discrimination [5]. Specifically, we extend these prior findings by showing that regional far-right political support is positively associated with immigrants' number of work-related performance errors, thus implying that their chances of attaining (and maintaining) the financial stability that employment offers might be negatively affected. This stability is further threatened by our finding that natives, in contrast, exhibit fewer performance errors in regions with higher far-right political support. To our knowledge, this study is the first demonstrating relationships between regional far-right political support and performance-related work outcomes.

Finally, this work introduces professional sports as a viable context in which to study immigrants in real-world, organizational settings using objective measures of performance-related work outcomes (e.g., performance errors). Although there are likely relatively few immigrants working in the majority of organizations [7], professional sports teams are much more diverse as it relates to athletes' immigration status– 23% of basketball players in our sample were immigrants–and likely to become increasingly diverse with time [68]. With this study, we hope to encourage fellow researchers investigating immigrants as a subgroup of the working population to consider the unique benefits that the professional sports context has to offer (e.g., large, openly-available datasets of individuals nested within teams) when proposing and testing theoretical models.

## Practical implications

We demonstrate that the political environment in which immigrants reside is negatively associated with the performance-related work outcomes of even the most successful and capable immigrants in their field. Our findings are therefore practically relevant for organizations as they could be interpreted as surmising that there is no level of success that immigrants can attain that will protect them from the corrosive influence of a threatening, external political environment. Thus, organizational leaders would do well to try and insulate their organization from the potentially divisive influence of the external political situation by, for example, prohibiting employees from wearing discriminatory or politically-laced apparel (e.g., MAGA hats) and creating inclusive climates [69] as an internal buffer against immigrant stereotype threats. Concurrently, team leaders could publicly recognize the unique added value of each of their (immigrant) employees to reassure them of their belongingness within the organization and profession as a whole.

While our research took place in a specific professional sport setting, the main implications might also be transferable to more traditional jobs. For example, similar to professional athletes, professional drivers of busses, taxis and limousines are often immigrants [70] and must perform their job tasks under the watchful eyes of community members. Furthermore, as driving also involves well-learned and automatic tasks, immigrant drivers may be at risk of the explicit monitoring associated with stereotype threat. Moreover, and similar to professional sports teams, leading (IT) startups in the U.S. rely heavily on recruiting the best talents worldwide. In 2018, for example, a study showed that 55 percent of the startup companies in the U.S. valued $1 billion or more were founded by immigrants [71]. If executives and other top leaders make more errors as a result of anti-immigrant threats in their environment, that could seriously harm the organizational performance and even the overall economic performance in the U.S. Finally, our findings likely also pertain to international companies who send their employees for assignments abroad, as the political environment may explain in part why these employees frequently do not perform to their supervisors' expectations [72].

## Limitations and future directions

Despite the strengths of our study, it is not without limitations. First, the research context of our study is the NBA, an elite professional organization comprised of athletes who are not representative of the population as a whole. In particular, and relevant to the psychological mechanism we propose drives our results, NBA players receive status and public exposure (including scrutiny) far above what the average immigrant (or non-immigrant) does. These factors are important to consider given that perceptions of status and exposure to outgroup members can affect one's experience of stereotype threat [46] and performance [73]. Immigrant workers in less public roles may, therefore, be at lower risk of stereotype threat arising from the region in which they work; though research has also shown stereotype threat to negatively affect performance when alone [74]. Future research can build on our model, as well as address the limited external validity of our unique study sample, by extending our findings to organizational contexts that better represent the general population of employed individuals and their exposure to public scrutiny (or lack thereof) while working. Examples of this could include studying performance differences between immigrant and native workers in roles with varying amounts of public exposure (e.g., office workers vs. postal workers vs. news reporters) across regions with varying far-right political support. Such work would help provide external validity to our findings by replicating them in more representative samples of workers. That being said, the fact that we nonetheless find our results in an elite group of immigrant professionals underlines the impact that regional far-right political support can have both on significantly less-famous immigrants (e.g., expatriates) and significantly more vulnerable immigrants (e.g., refugees). Second, our findings are limited by the fact that they are drawn from a young sample of employees. Although our results suggest that younger immigrant employees can be negatively affected by regional far-right political support, older immigrant employees may become desensitized to it due to inurement (i.e., habituation) effects [75] or personality changes over time (e.g., self-confidence and independence increase in middle age) [76]. On the other hand, older native employees may experience greater stereotype lift effects than their younger counterparts simply from having been exposed to more anti-immigrant stereotypes over their lifetime. To address this limitation, future research could build on our theoretical model by testing age as a potential moderator of interest in an age-diverse sample of employees. Third, our operationalization of regional far-right political support is based on state-level election data. Although this state-level measure may be considered coarse, prior research has found inter-state differences in personality (e.g., openness to experience, liberal values) to vary

systematically [77,78] and that people can accurately report differences in residents' personality characteristics across regions [79,80]. Specific to our study, Republican-leaning states are often referred to as conservative and close-minded [78], which is in line with findings that Republicans self-report more negative perceptions of immigrants (e.g., they are not to be trusted) than do Democrats or Independents [81]. This prior work supports our assumption that differences in attitudes towards immigrants exist at the state level and that these attitudes are salient to immigrants. Furthermore, and lending credence to our methodology, U.S. state-level election results have recently been shown to predict employees' integration into their organization [82] as well as organizations' corporate performance [83]. Finally, although we propose chronic stereotype threat and stereotype lift as the psychological mechanisms responsible for players' performance errors, we could not directly test this with our dataset. Moreover, the observational nature of the data prevent causal conclusons from being made and experimental studies are needed to determine if, and how, immigrant status causes work-related performance errors in regions with high far-right political support. In the future, researchers could employ mixed method designs matching basketball statistics with player interviews to better understand players' personal experiences of stereotype threat and stereotype lift before games. This could help elucidate not only the underlying mechanism(s) driving these individual effects (e.g., social categorization salience resulting from political rhetoric on immigration), but also the role of potential moderators such as perceived inter-group conflict and, relatedly, prejudice [84]. Qualitative studies building on our theoretical model could thereby provide more nuanced insight into which individuals are particularly susceptible to stereotype threat/lift effects, and why. Although mixed method designs increases the complexity of one's study procedures, initial work has demonstrated the predictive validity of such approaches [85].

## Conclusions

Addressing calls for further research on immigrants as a demographic group [7,15], as well as the importance of the external political context on work-relevant outcomes [20,21], our study investigates the relationship between regional far-right political support, employees' immigrant status, and their work performance. Specifically, we demonstrate the positive link between regional far-right political support and performance errors in immigrants (regardless of their skin color). This raises the possibility that the external political environment can result in negative consequences for immigrants and the organizations for whom they work. Moreover, this study demonstrates the utility of using professional sports data to explore immigrants' work-related outcomes.

## Acknowledgments

We thank the Konstanz Future of Work Lab's team members for valuable feedback on this paper during multiple research workshops. Additionally, we thank our research assistants Ali Zarabi and Theo Kaiser for their help coding NBA players' skin color.

## Author Contributions

**Conceptualization:** Benjamin A. Korman.

**Data curation:** Benjamin A. Korman.

**Formal analysis:** Benjamin A. Korman.

**Funding acquisition:** Florian Kunze.

**Supervision:** Florian Kunze.

**Writing – original draft:** Benjamin A. Korman.

**Writing – review & editing:** Benjamin A. Korman, Florian Kunze.

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
