## [Decision Letter · Decision Letter 0]

18 Apr 2023

PONE-D-23-04965Political context and immigrants’ work-related performance errors: Insights from the National Basketball AssociationPLOS ONE

Dear Dr. Korman,

Thank you for submitting your manuscript to PLOS ONE. After careful consideration, we feel that it has merit but does not fully meet PLOS ONE’s publication criteria as it currently stands. Therefore, we invite you to submit a revised version of the manuscript that addresses the points raised during the review process.

Your manuscript has been assessed by an expert reviewer, whose comments are appended below. As you will see from the full report, the reviewer has highlighted some areas where your manuscript would benefit from additional discussion and a stronger theoretical underpinning. Please ensure you respond to each point carefully in your response to reviewers document, and modify your manuscript accordingly. Please note that we have only been able to secure a single reviewer to assess your manuscript. We are issuing a decision on your manuscript at this point to prevent further delays in the evaluation of your manuscript. Please be aware that the editor who handles your revised manuscript might find it necessary to invite additional reviewers to assess this work once the revised manuscript is submitted. However, we will aim to proceed on the basis of this single review if possible.

We look forward to receiving your revised manuscript.

Kind regards,

Dr Joseph Donlan

Senior Editor

PLOS ONE

2. You indicated that ethical approval was not necessary for your study. We understand that the framework for ethical oversight requirements for studies of this type may differ depending on the setting and we would appreciate some further clarification regarding your research. Could you please provide further details on why your study is exempt from the need for approval and confirmation from your institutional review board or research ethics committee (e.g., in the form of a letter or email correspondence) that ethics review was not necessary for this study? Please include a copy of the correspondence as an ""Other"" file.

Reviewers' comments:

Reviewer's Responses to Questions

**Comments to the Author**

1. Is the manuscript technically sound, and do the data support the conclusions?

Reviewer #1: Yes

2. Has the statistical analysis been performed appropriately and rigorously? 

Reviewer #1: Yes

3. Have the authors made all data underlying the findings in their manuscript fully available?

Reviewer #1: Yes

4. Is the manuscript presented in an intelligible fashion and written in standard English?

Reviewer #1: Yes

5. Review Comments to the Author

Reviewer #1: The research conducted is both original and rigorous, and it addresses an important current issue. The variables are well-defined and the use of objective field data is a strength. However, my main concern is that the authors do not delve deeply into the impact on native players. Do the authors believe that the right-wing anti-immigration discourse is responsible for both expected effects, as described in H1 and H2, by highlighting the inter-group conflict between "natives" and "immigrants"? If this is the case, then the effects of social categorization (as described by Tajfel and Turner, among others) should be considered in the chapter on "theory and hypotheses," as well as in the discussion. In my opinion, the salience of these social categories and the subsequent conflict between the two groups is the necessary precondition for the occurrence of the Stereotype Lift effect, particularly in the authors' study.

Another theoretical issue that I believe requires further nuance in the discussion is the significant difference between the work context of an NBA player (or other sports with a mass audience) and any other work organization. From a psychosocial perspective, the public exposure of these "workers" in the NBA is a variable that should be considered more rigorously in the interpretation of the results. The authors should elaborate more on this aspect, as it does not invalidate their main findings but would add more value to the article. "The public exposure of these "workers" in the NBA, which distinguishes them from general workers, should be considered more carefully in the limitations section or as a topic for future work."

6. PLOS authors have the option to publish the peer review history of their article (what does this mean?). If published, this will include your full peer review and any attached files.

Reviewer #1: **Yes: **Berta Chulvi Ferriols

---

## [Author Response · Author response to Decision Letter 0]

16 May 2023

To Whom It May Concern,

We have uploaded our responses to the reviewer and editor comments together with the other revision documents (manuscript, figures, etc.). Our responses are available in the document titled "Response to Reviewers.docx". Please let us know should this not suffice.

Kind regards

The Authors

---

## [Decision Letter · Decision Letter 1]

2 Aug 2023

PONE-D-23-04965R1Political context and immigrants’ work-related performance errors: Insights from the National Basketball AssociationPLOS ONE

Dear Dr. Korman,

Thank you for submitting your manuscript to PLOS ONE. As you aware, some additional concerns were noted with your manuscript at a late stage. Please note that as per our publication criteria, PLOS ONE requires that all experiments, statistics and other analyses are performed to a high technical standard, described in sufficient detail and adhere to appropriate reporting guidelines and community standards. Conclusions must be presented in an appropriate fashion and be supported by the data (Please see http://journals.plos.org/plosone/s/criteria-for-publication).

After careful consideration, we feel that it has merit but does not fully meet PLOS ONE’s publication criteria as it currently stands. Therefore, we invite you to submit a revised version of the manuscript that addresses the points raised below.

Kind regards,

George Vousden

Deputy Editor in Chief

PLOS ONE

*On behalf of,*

Chunyu Zhang

Academic Editor

PLOS ONE

We look forward to receiving your revised manuscript.

Kind regards,

Chunyu Zhang

Academic Editor

PLOS ONE

Journal Requirements:

Additional Editor Comments:

The submitted manuscript is observational in nature. As such, it is not appropriate to make draw conclusions related to causation. The following fragments must be revised to indicate that an association between the variables of note was reported, but must not imply that the results demonstrate causation or direction of a given effect. You may speculate on the potential for causation in your manuscript, but not suggest your results are sufficient to demonstrate this.

Line number below refer to the clean version of the manuscript in Editorial Manager.

Line 80-82: “can be predisposed” should be revised (e.g. with “show increased performance errors”)

Line 88-90: This statement implies causation and must be revised.

Line 96-99: This statement implies causation and must be revised.

Line 336-339: This statement implies causation and must be revised.

Line 353-356: Whilst most of the text in lines 353-355 is acceptable, “jeopardizing” implies causation and should be revised.

Line 372-374: This statement implies causation and must be revised.

Line 374:377: This statement implies causation and must be revised.

Limitations section: Please extend the discussion beginning on line 442 to include how the observational nature of the data prevents conclusions related to causation being made.

Line 460-462: the text beginning ‘thus’ should be toned down. Whilst you may speculate on the causes of the patterns observed here, this sentence overstates your conclusions. It would be acceptable to indicate as follows:

“Specifically, we demonstrate the positive link between regional far-right political support and performance errors in immigrants (regardless of their skin color). This raises the possibility that the external political environment can result in negative consequence for immigrants and the organizations for whom they work.”

In addition to the above, please also amend your Methods section to provide further detail on how immigrant status was determined. The specific data sources used to determine this should be required. In addition, it is stated that the approach was “similar to other research [29]”. Please supplement this reference with references to original research studies using the same approach as described here. 

Reviewers' comments:

Reviewer's Responses to Questions

**Comments to the Author**

1. If the authors have adequately addressed your comments raised in a previous round of review and you feel that this manuscript is now acceptable for publication, you may indicate that here to bypass the “Comments to the Author” section, enter your conflict of interest statement in the “Confidential to Editor” section, and submit your "Accept" recommendation.

Reviewer #1: All comments have been addressed

Reviewer #2: All comments have been addressed

2. Is the manuscript technically sound, and do the data support the conclusions?

Reviewer #1: Yes

Reviewer #2: Yes

3. Has the statistical analysis been performed appropriately and rigorously? 

Reviewer #1: Yes

Reviewer #2: Yes

4. Have the authors made all data underlying the findings in their manuscript fully available?

Reviewer #1: Yes

Reviewer #2: Yes

5. Is the manuscript presented in an intelligible fashion and written in standard English?

Reviewer #1: Yes

Reviewer #2: Yes

6. Review Comments to the Author

Reviewer #1: I am pleased to say that the authors have taken into account the majority of my suggestions. They thoroughly considered and incorporated nearly all of my suggestions in the limitation sections, and I think the paper could be published in the actual form.

Reviewer #2: (No Response)

7. PLOS authors have the option to publish the peer review history of their article (what does this mean?). If published, this will include your full peer review and any attached files.

Reviewer #1: **Yes: **Berta Chulvi Ferriols

Reviewer #2: No

---

## [Author Response · Author response to Decision Letter 1]

10 Aug 2023

Dear Dr. Vousden and Dr. Zhang,

Thank you for giving us another opportunity to revise our manuscript and bring it in line with PLOS ONE reporting standards. Below we address your comments, with each of our responses presented beginning with Res: and ending with //.

The submitted manuscript is observational in nature. As such, it is not appropriate to make draw conclusions related to causation. The following fragments must be revised to indicate that an association between the variables of note was reported, but must not imply that the results demonstrate causation or direction of a given effect. You may speculate on the potential for causation in your manuscript, but not suggest your results are sufficient to demonstrate this.

Res:

We understand the importance of emphasizing that our findings are correlational in nature and appreciate your detailed insight on which parts of the text require changing and how they can be effectively changed.

//

Line number below refer to the clean version of the manuscript in Editorial Manager.

Line 80-82: “can be predisposed” should be revised (e.g. with “show increased performance errors”)

Res:

This section of text has been replaced with the text you proposed.

//

Line 88-90: This statement implies causation and must be revised.

Res:

This has been revised.

//

Line 96-99: This statement implies causation and must be revised.

Res:

This has been revised.

//

Line 336-339: This statement implies causation and must be revised.

Res:

This has been revised.

//

Line 353-356: Whilst most of the text in lines 353-355 is acceptable, “jeopardizing” implies causation and should be revised.

Res:

This has been revised.

//

Line 372-374: This statement implies causation and must be revised.

Res:

This has been revised.

//

Line 374:377: This statement implies causation and must be revised.

Res:

This has been revised.

//

Limitations section: Please extend the discussion beginning on line 442 to include how the observational nature of the data prevents conclusions related to causation being made.

Res:

As you request, we now mention that the observational nature of our data preclude causal claims and that future studies are needed to establish causality.

//

Line 460-462: the text beginning ‘thus’ should be toned down. Whilst you may speculate on the causes of the patterns observed here, this sentence overstates your conclusions. It would be acceptable to indicate as follows:

“Specifically, we demonstrate the positive link between regional far-right political support and performance errors in immigrants (regardless of their skin color). This raises the possibility that the external political environment can result in negative consequence for immigrants and the organizations for whom they work.”

Res:

This has been revised. Thank you for providing an adequate alternative to what we originally wrote, this has eased the revision process.

//

In addition to the above, please also amend your Methods section to provide further detail on how immigrant status was determined. The specific data sources used to determine this should be required. In addition, it is stated that the approach was “similar to other research [29]”. Please supplement this reference with references to original research studies using the same approach as described here.

Res:

As you request, we now specify where exactly the information on immigrant status came from and cite previous research studies operationalizing immigrant status based on study participants’ country of birth.

We hope you find the changes in this revision satisfactory but stand ready to make additional alterations if necessary.

//

---

## [Decision Letter · Decision Letter 2]

26 Sep 2023

Political context and immigrants’ work-related performance errors: Insights from the National Basketball Association

PONE-D-23-04965R2

Dear Dr. Korman,

We’re pleased to inform you that your manuscript has been judged scientifically suitable for publication and will be formally accepted for publication once it meets all outstanding technical requirements. Within one week, you’ll receive an e-mail detailing the required amendments. When these have been addressed, you’ll receive a formal acceptance letter and your manuscript will be scheduled for publication.

In the meantime, I would like you to make the final changes in line with the minor comments of Reviewer #1. Although minor, I think they will present the timely aspects of your manuscript.  

Kind regards,

Cengiz Erisen

Academic Editor

PLOS ONE

Additional Editor Comments (optional):

Reviewers' comments:

Reviewer's Responses to Questions

**Comments to the Author**

1. If the authors have adequately addressed your comments raised in a previous round of review and you feel that this manuscript is now acceptable for publication, you may indicate that here to bypass the “Comments to the Author” section, enter your conflict of interest statement in the “Confidential to Editor” section, and submit your "Accept" recommendation.

Reviewer #1: All comments have been addressed

Reviewer #2: All comments have been addressed

2. Is the manuscript technically sound, and do the data support the conclusions?

Reviewer #1: Yes

Reviewer #2: Yes

3. Has the statistical analysis been performed appropriately and rigorously? 

Reviewer #1: Yes

Reviewer #2: Yes

4. Have the authors made all data underlying the findings in their manuscript fully available?

Reviewer #1: No

Reviewer #2: Yes

5. Is the manuscript presented in an intelligible fashion and written in standard English?

Reviewer #1: Yes

Reviewer #2: Yes

6. Review Comments to the Author

Reviewer #1: Dear authors

I only realised it on this last reading but the first sentence in the abstract assumes that the rise of Far-right parties is produced by the increase of International Immigration. There is no empirical support for this assumption quoted in the paper. The development of Far-right success at this moment in Europe is a complex phenomenon with multiple causes and I would propose to start the abstract with a sentence like this one: “Support for far-right parties in Europe has grown over the last decade. One of the arguments most often used by these parties is the need to control international immigration.”

Or other similar that works well by introducing the article but avoid this type of theoretical assumptions.

On another note, I think the article is very timely at this time after the controversy unleashed by the conduct of the president of the Spanish Football Federation (Mister Luis Rubiales) and the world champion, Jenny Hermoso. And one RQ is how the reactions supporting for the president, which are also occurring, will affect the football of the players. Perhaps you could think of an extension of the work to deal with sexist attitudes?

Reviewer #2: (No Response)

7. PLOS authors have the option to publish the peer review history of their article (what does this mean?). If published, this will include your full peer review and any attached files.

Reviewer #1: **Yes: **Berta Chulvi

Reviewer #2: No

---

## [Editor Report · Acceptance letter]

12 Jul 2023

PONE-D-23-04965R1 

Political context and immigrants’ work-related performance errors: Insights from the National Basketball Association 

Dear Dr. Korman:

I'm pleased to inform you that your manuscript has been deemed suitable for publication in PLOS ONE. Congratulations! Your manuscript is now with our production department. 

Kind regards, 

on behalf of

Dr. Chunyu Zhang 

Academic Editor

PLOS ONE